# Accelerated Quantitative 3D UTE-Cones Imaging Using Compressed Sensing

**DOI:** 10.3390/s22197459

**Published:** 2022-10-01

**Authors:** Jiyo S. Athertya, Yajun Ma, Amir Masoud Afsahi, Alecio F. Lombardi, Dina Moazamian, Saeed Jerban, Sam Sedaghat, Hyungseok Jang

**Affiliations:** Department of Radiology, University of California, San Diego, CA 92103, USA

**Keywords:** compressed sensing, UTE, quantitative MRI, image reconstruction

## Abstract

In this study, the feasibility of accelerated quantitative Ultrashort Echo Time Cones (qUTE-Cones) imaging with compressed sensing (CS) reconstruction is investigated. qUTE-Cones sequences for variable flip angle-based UTE T_1_ mapping, UTE adiabatic T_1ρ_ mapping, and UTE quantitative magnetization transfer modeling of macromolecular fraction (MMF) were implemented on a clinical 3T MR system. Twenty healthy volunteers were recruited and underwent whole-knee MRI using qUTE-Cones sequences. The k-space data were retrospectively undersampled with different undersampling rates. The undersampled qUTE-Cones data were reconstructed using both zero-filling and CS reconstruction. Using CS-reconstructed UTE images, various parameters were estimated in 10 different regions of interests (ROIs) in tendons, ligaments, menisci, and cartilage. Structural similarity, percentage error, and Pearson’s correlation were calculated to assess the performance. Dramatically reduced streaking artifacts and improved SSIM were observed in UTE images from CS reconstruction. A mean SSIM of ~0.90 was achieved for all CS-reconstructed images. Percentage errors between fully sampled and undersampled CS-reconstructed images were below 5% for up to 50% undersampling (i.e., 2× acceleration). High linear correlation was observed (>0.95) for all qUTE parameters estimated in all subjects. CS-based reconstruction combined with efficient Cones trajectory is expected to achieve a clinically feasible scan time for qUTE imaging.

## 1. Introduction

MRI has been an important diagnostic imaging modality, owing to its noninvasiveness and lack of risk of exposure to ionizing radiation while providing excellent visualization of anatomical structures and physiological functions. However, the major limitation encountered in clinics is a relatively longer scan time than other imaging modalities such as CT, X-ray, and ultrasound. MRI requires that patients remain still during scanning to avoid motion that can cause imaging artifacts and misregistration between images. Moreover, prolonged MR exams may be impractical for pediatric patients or those in pain. Therefore, accelerated MR acquisition is crucial to minimizing patient discomfort by reducing stationary periods. In addition, accelerated MRI scans may directly help reduce healthcare costs by allowing for higher patient throughput.

Despite achieving excellent soft tissue contrast, clinical MRI has a limitation in imaging short T_2_ tissues such as tendons, cortical bone, ligaments, and menisci. Due to their rapid signal decay over free induction decay (FID), these anatomical structures remain undetected with conventional MR pulse sequences that have long echo times (TEs). Ultrashort echo time (UTE) MR sequences allow for direct imaging of tissues with short T_2_ values as they utilize significantly shortened TEs (on the order of microseconds), which cannot be achieved by clinical MR sequences (with TEs on the order of milliseconds) [1,2,3,4,5]. Three-dimensional UTE imaging typically utilizes a non-Cartesian center-out radial encoding scheme in which a 3D spherical k-space is encoded with rotating half-radial projection lines. A more efficient, 3D spiral Cones trajectory has been incorporated in UTE imaging to encode a 3D k-space with reduced acquisition times (~2.5×—5× acceleration) [6], more suitable for in vivo imaging [7,8]. In addition, the 3D Cones trajectory possesses an innate property to oversample the low-frequency region of a k-space, which can provide autocalibration for parallel imaging (PI) to achieve additional accelerations [9,10].

Recently, quantitative UTE (qUTE) imaging techniques combined with the efficient 3D Cones trajectory, such as UTE-based T_2_* (UTE-T_2_*) mapping [11], UTE-based T_1_ (UTE-T_1_) mapping [7], UTE-based adiabatic T_1ρ_ (UTE-Adiab-T_1ρ_) mapping [12], and UTE-based quantitative magnetization transfer (UTE-qMT) modeling [13], have surfaced as new potential biomarkers to characterize musculoskeletal (MSK) tissues with short T_2_ values such as deep cartilage, menisci, ligaments, tendons, and bone. Variable TE-based UTE-T_2_* mapping and variable flip angle (VFA)-based UTE-T_1_ mapping have demonstrated quantification of T_2_* and T_1_ relaxation times of both short and long T_2_ tissues in the knee joint [7]. The UTE-Adiab-T_1ρ_ mapping technique has shown efficacy at assessing changes in proteoglycan and collagen contents in various knee joint tissues, with dramatically reduced sensitivity to the magic angle effect [14]. UTE-qMT modeling has demonstrated the assessment of macromolecular fraction (MMF) [15,16]. Despite the promises with the qUTE-Cones imaging techniques in the MSK system, the total acquisition time still remains long, since multiple images with different contrasts (i.e., different imaging parameters) need to be acquired [17,18,19,20]—an issue that poses a major impediment to translating these techniques into clinical practices.

Compressed sensing (CS) theoretically allows for the recovery of unknown sparse signals from underdetermined measurements [21]. With CS, MR images in a sparse representation can be reconstructed from randomly undersampled k-space data with the help of appropriate nonlinear optimization algorithms. Since typical MR images are not sparse in the native spatial domain, a sparsifying transform such as discrete wavelet transform (DWT) is often utilized [22]. Recently, parallel imaging (PI) has been combined with CS techniques to achieve artifact-free image reconstruction with even higher acceleration [10]. In this first-of-its-kind, previously unpublished study, the feasibility and efficacy of PI-based CS are explored in 3D qUTE-Cones imaging. The data are undersampled at different levels, which are then reconstructed into images with highly acceptable imaging standards, allowing for error-free and reliable qUTE parameter mapping along with reduced acquisition time.

### Theory

The theory behind compressed sensing stems from the idea to intentionally violate the Nyquist sampling rate by acquiring a lower number of measurements. In MRI, a desired MR image is reconstructed by calculating the inverse discrete Fourier transform of fully sampled k-space data. Undersampling of the input k-space data reduces acquisition time at the cost of aliasing artifacts that occur in the spatial domain due to Nyquist violation in k-space (i.e., reduced field of view (FOV)) [23]. CS in MRI utilizes a randomized undersampling scheme to ensure that the aliasing artifact is noise-like (i.e., incoherent) in either the native spatial domain or DWT domain. The generic CS MR inverse problem can be given as:(1)$$\hat{x}=argmin_{x}\frac{1}{2}\|Ax-y{\|}_{2}^{2}+R\left\{x\right\},$$
where *x* is the complex image, *y* is k-space data that are undersampled, and *R* is a regularization parameter. The sensing matrix *A* is composed as:(2)A=PFS,
where *P* denotes a sampling operator, *F* denotes a discrete Fourier transform operator, and *S* denotes coil sensitivity information [24]. The first term in the right-hand side of Equation (1) ensures data fidelity, and the second term promotes sparsity in a certain domain (e.g., DWT domain). For accelerated MRI with undersampled k-space data, the regularization term, *R*, in Equation (1) may be critical to suppress the noise-like aliasing artifact, which typically utilizes gradient information in the image domain, such as total variation (i.e., finite difference). In literature, DWT-based CS combined with PI has proven to be successful in MRI reconstruction [25,26]. By employing DWT as a sparsifying transform, the data representation becomes much sparser, thereby enabling CS to achieve its full efficiency.

## 2. Materials and Methods

### 2.1. Pulse Sequence

The UTE technique allows for direct acquisition of an FID signal with short T_2_ values lasting only a few hundreds of microseconds after radiofrequency (RF) excitation, before major T_2_* decay has occurred. Figure 1a shows a pulse sequence diagram of 3D UTE-Cones sequence used for imaging in this study. The 3D UTE-Cones sequence employs a short rectangular radiofrequency (RF) pulse for nonselective excitation of spins, followed by data acquisition with efficient 3D spiral Cones trajectories. For the 3D Cones imaging, a 3D spiral arm is first generated [27] for each Cone (e.g., for ~100–300 total polar angles to cover the 3D k-space through kz-axis). Then, the spiral arm is rotated around the kz-axis to generate different spiral arms and hence cover the kx–ky plane in each Cone, yielding ~10,000–40,000 spokes to encode the 3D spherical k-space. UTE-Adiab-T_1ρ_ [12] and UTE-qMT [15] can be achieved by applying an additional pulse sequence module designed to achieve the desired signal contrast in front of the UTE-Cones sequence. Figure 1b,c show the signal preparation module used for UTE-Adiab-T_1ρ_ and UTE-qMT imaging, respectively. The 3D Adiab-T_1ρ_ UTE-Cones sequence used a train of adiabatic full-passage pulses to generate T_1ρ_ contrast, followed by 3D UTE-Cones data acquisition. For UTE-qMT, a Fermi pulse was used for MT preparation. After the pulse preparation, multiple spokes were sampled to speed up data acquisition (N_sp_: number of spokes per preparation). Figure 1d illustrates a simplified example of the 3D Cones-based k-space trajectory.

### 2.2. Recruitment

In this study, 20 healthy volunteers were recruited for knee MRI following human study guidelines issued by our Institutional Review Board. Average age of the included volunteers was 38 years, with range between 24 and 60 years of age. The cohort included 7 male and 13 female subjects. Additionally, a female subject aged 27 years with a previous anterior cruciate ligament injury was recruited as a pathologic case to demonstrate the feasibility in extending the technique to abnormal dataset as well.

### 2.3. Imaging Parameters

MRI was performed in a clinical 3T MRI scanner (MR750, GE Healthcare, Milwaukee, WI, USA). An 8-channel knee coil was used for both RF transmission and signal reception. In UTE-Cones imaging, both RF and gradient spoiling were performed to remove the remaining transverse magnetization after each data acquisition. The 3D UTE-Cones sequences, which were previously reported in [7,12,15], were used for qUTE-Cones imaging. The k-space data were acquired with 23,869 spokes in a full Nyquist sampling rate using the imaging parameters listed in Table 1.

### 2.4. Compressed Sensing-Based Image Reconstruction

The k-space data were retrospectively undersampled at four different levels (i.e., 25%, 35%, 50%, and 65%) using a pseudorandom bit-reversal ordering scheme. This was followed by iterative density compensation [29], which is crucial due to the non-Cartesian nature of Cones trajectory that does not possess uniform distribution of data points. Coil sensitivity was estimated using the complex image reconstructed with zero-filling in each channel using nonuniform FFT [30].

The objective function of CS reconstruction based-based on *l*_1_ norm and *l*_2_ norm was posed as below:(3)$$|PFSx-y{|}_{2}^{2}+\lambda |\phi x{|}_{1},$$
where *F* is Fourier transform, *P* is a sampling operator, *S* is coil sensitivity, *x* is an image to be reconstructed, *y* is k-space data, λ is a regularization parameter, and ϕ is discrete wavelet transform operator.

Berkeley Advanced Reconstruction Toolbox (BART) [31] was used to perform DWT-based CS reconstruction. The regularization parameter, λ, was empirically optimized to accommodate the current application and remained constant for the control group. The reconstructed images were input to the subsequent qUTE parameter quantification process. Figure 1e shows a complete block diagram representation for the entire flow.

### 2.5. Parameter Fitting

Data analysis was performed using MATLAB (The MathWorks, Natick, MA, USA). Ten different regions of interest (ROIs) (i.e., anterior femoral cartilage (AFC), medial femoral cartilage (MFC), posterior femoral cartilage (PFC), patellar cartilage (PC), tibial plateau cartilage (TPC), anterior lateral meniscus (ALM), posterior lateral meniscus (PLM), anterior cruciate ligament (ACL), posterior cruciate ligament (PCL), and patellar tendon (PT)) were drawn in the in vivo knee images from 20 healthy subjects. Levenberg–Marquardt algorithm was employed for quantification of pixelwise parameters in UTE-T_1_, UTE-Adiab-T_1ρ_ and UTE-qMT within the manually drawn ROIs [7,12,13], and their mean and standard deviation were computed.

### 2.6. Data Analysis

Structural similarity (SSIM) was utilized to assess the reconstruction performance for morphological UTE-Cones imaging [32]. The SSIM combines three different components to yield the overall metric that includes luminance, contrast, and structure. A mean SSIM (MSSIM) index evaluates the overall image quality between completely sampled and partially sampled reconstruction for various quantification parameters, as given by the following equation:(4)$$\mathrm{MSSIM}\left(A,B\right)=\frac{1}{M}\sum_{j=1}^{M}\mathrm{SSIM}(a_{j},b_{j}),$$
where *A* is the reference (fully sampled reconstructed image) and *B* is the CS-reconstructed image, respectively; $a_{j}$ and $b_{j}$ are the image contents at the *j*th local window; and M is the number of local windows in the image. *A* mean of MSSIM was calculated from the entire sample set for describing the achieved reconstruction performance.

Pearson’s correlation was calculated between qUTE parameters estimated with fully sampled and undersampled data, based on the mean values from all ROIs in all subjects (i.e., 20 subjects × 10 ROIs = 200 data points).

## 3. Results

### 3.1. Morphological Assessment

CS reconstruction was compared with zero-filled reconstruction at various undersampling levels (i.e., 25, 35, 50, and 65%). In all 20 subjects, CS provided a discernible morphological improvement of the reconstructed images while zero-filled reconstruction showed pronounced streaking artifacts at high undersampling rates.

In Figure 2, the UTE–T_1_ images reconstructed at different undersampling rates using zero-filling or CS are shown. At 25% undersampling, the images were visually similar to those with fully sampled data. At 35% undersampling, a mild intensity change and contrast variation were pictured throughout the sagittal slice of femur in the zero-filled reconstruction, while such changes were hardly visible using CS-based reconstruction (yellow arrows). At 50% and 65% undersampling, both CS and zero-filling introduced streaking, but CS showed improved image quality with higher details. Similar tendencies were observed in UTE-Adiab-T_1ρ_ and UTE-qMT. In UTE-Adiab-T_1ρ_ (Figure 3), streak artifacts were manifest in the tibial bone (yellow arrows) at 35%, 50%, and 65% undersampling rates for the zero-filled reconstruction. In particular, at a high undersampling rate (i.e., 65%) the images reconstructed with zero-filling showed blurred, compromised high-frequency details, while images with CS showed much-improved details. As seen in Figure 4, UTE-MT images obtained at a 50% undersampling rate using CS performed much better than the images reconstructed using the zero-filling method. Artifacts were consistently exhibited at higher undersampling rates in the zero-filled reconstruction as opposed to the CS reconstruction.

The MSSIMs from the zero-filled reconstruction and CS reconstruction with the total dataset of 20 subjects are presented in Table 2. In all qUTE techniques, the MSSIM dropped with an increased undersampling rate due to exacerbated reconstruction error. For most cases, the MSSIM with the CS-based reconstruction was over 0.9 with up to 50% undersampling while the zero-filled reconstruction recorded much lower value (~0.77).

### 3.2. qUTE Parameter Mapping

To illustrate the pixelwise mapping of T_1_, T_1ρ_, and MMF parameters, four ROIs representing four different tissue types (i.e., PT, AFC, PCL, and PLM) out of a total of 10 ROIs were selected. The results of quantitative parameter mapping in the selected ROIs are visualized in Figure 5, Figure 6 and Figure 7. Figure 5 shows the results of UTE-T_1_ mapping at various undersampling levels where the estimated parameters fall under the admissible range, as reported in [4]. Figure 6 and Figure 7 display the results of UTE-Adiab-T_1ρ_ and UTE-qMT mapping at different undersampling levels.

### 3.3. Quantitative Assessment of CS-Based qUTE Parameters

Table 3 provides the mean and standard deviation of all qUTE parameters estimated in all tissue ROIs at different undersampling levels, as well as the corresponding mean percentage errors with respect to the reference values obtained from fully sampled data. The estimated parameters from CS-reconstructed images exhibited a percentage error below 5% for up to 50% undersampling in most of ROIs. PCL and PLM exhibited relatively higher errors, presumably due to the high partial volume effect and the small size of ROI, which is more susceptible to the reconstruction errors.

Figure 8 depicts the scatter plots of mean qUTE parameters estimated in all 200 ROIs from all subjects, as well as the Pearson’s correlation between fully sampled and undersampled CS with various undersampling levels. Each ROI for each tissue type is coded with a different color to identify the outlier points. The high degree of linearity indicates that the quantitative parameters from undersampled data (y-axis in scatter plots in Figure 8) correlate well with the corresponding parameters from fully sampled data (x-axis in scatter plots in Figure 8), despite the reduced data. Among all ROIs, qUTE parameters estimated in PCL (orange dots) included many outliers due to the abovementioned reasons (i.e., partial volume effect and small ROI).

### 3.4. CS Reconstruction from a Degenerated Knee

Figure 9 portrays a special case with a 27-year-old female patient who has tibial tunnel from a previous ACL reconstruction, which is clearly depicted in the CS-reconstructed images (red arrows). Susceptibility artifacts are shown in the tibial tunnel (red arrows) on the distal portion, as shown in the sagittal UTE MRI data of the knee. Similar to the healthy control dataset, the presence of artifacts and blurring of information is visible in various sections of images from a zero-filled reconstruction. CS reconstruction, on the other hand, provides a consistent and reliable image quality up until 65% undersampling (yellow arrows).

## 4. Discussion

In this work, the feasibility of accelerated qUTE-Cones imaging using the CS reconstruction method was explored for the first time to our best knowledge. The qUTE parameter mappings inherently require multiple repeated acquisitions of images to achieve different contrasts used in signal modeling and parameter fitting. For example, T_1ρ_ mapping requires the acquisition of multiple images with different TSLs, resulting in different degrees of T_1ρ_ weighting (i.e., based on exponential decays) to fit the relaxation parameter. This in turn imposes a longer scan time, which is especially problematic in in vivo applications. A rudimentary approach to reducing the image acquisition time of any mapping is to limit the number of sequences (i.e., data points in the parameter domain), which could compromise accuracy and precision of the quantitative parameter estimation. Hence, it is vital to develop an accelerated qUTE technique that can reduce an acquisition time while generating parameter maps without significantly compromising the accuracy.

In this study, morphological image quality and accuracy of quantification maps were assessed to show the effectiveness of CS reconstruction in three qUTE techniques. In the morphological assessment, CS achieved the robust reconstruction of UTE images with highly reduced reconstruction errors with 50% undersampling (MSSIM > ~0.9), as shown in Figure 2, Figure 3, Figure 4 and Figure 5. In the experiment with the quantification of pixelwise qUTE parameters, CS yielded a percentage error below 5% and Pearson’s linear correlation over ~0.95 with 50% undersampling in most of ROIs. Therefore, it is expected that CS will shorten the total acquisition time of the qUTE techniques by a factor of 2 without dramatic degradation of the reconstructed image quality and accuracy in qUTE parameter estimation. In the current qUTE imaging protocol, with the 50% undersampling, CS reconstruction is expected to shorten the scan time down to 3 min 4 s, 4 min 27 s, and 4 min 49 s for UTE-T_1_, UTE-Adiab-T_1ρ_, and UTE-qMT, respectively, which is more suitable to add in the clinical MSK MRI workflow.

Despite being a highly useful tool for rapid qUTE imaging, the downside of CS is the need for finding an optimal regularization parameter involved in the reconstruction process, which is typically performed manually, and thus is time-consuming. CS is based on an information theory that allows for data reconstruction from highly undersampled measurements. The application of CS for clinical scenarios requires the estimation of optimal regularization parameter λ (i.e., Lagrange multiplier in the objective function to minimize). In this work, we empirically identified the best-suited regularization parameter that was maintained the same for all the cases and for all different qUTE techniques, which yielded a decent quality of image reconstruction, as demonstrated in the experimental results. This implies that the regularization parameter tuning for CS reconstruction is insensitive to the image contrast in qUTE-Cones.

This study has a limitation that only 20 healthy volunteers were included, without the inclusion of a cohort of patients. Although we presented a special case of one patient with abnormality in the knee joint, the inclusion of more cases with different pathologies will show the clinical feasibility in the general population. In future studies, we propose to recruit more patients with osteoarthritis to evaluate the diagnostic power of CS-based qUTE-Cones imaging.

## 5. Conclusions

We have demonstrated the feasibility and efficacy of CS reconstruction for quantitative 3D UTE-Cones imaging of the knee. The results showed that images can be reconstructed from undersampled data with a highly reasonable imaging standard to allow for robust qUTE parameter mapping with reduced error.

## Figures and Tables

**Figure 1 sensors-22-07459-f001:**
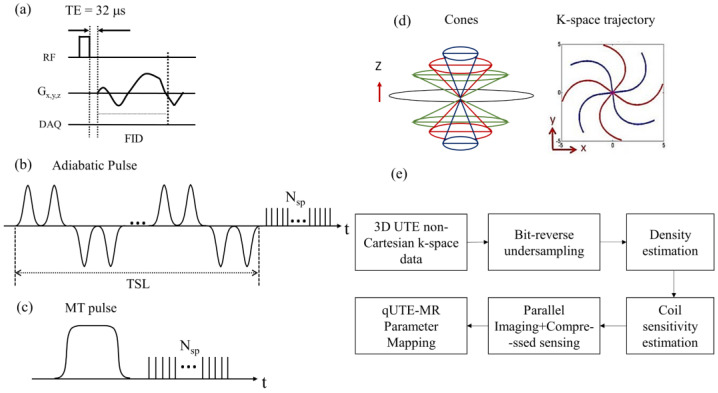
(**a**) UTE-Cones sequence with a short rectangular pulse, (**b**) adiabatic T_1__ρ_ signal preparation that uses a train of adiabatic full-passage pulses to generate T_1__ρ_ contrast followed by 3D UTE-Cones data acquisition, (**c**) MT preparation, (**d**) 3D Cones k-space trajectory, and (**e**) general flow involved in the accelerated quantitative UTE-Cones imaging.

**Figure 2 sensors-22-07459-f002:**
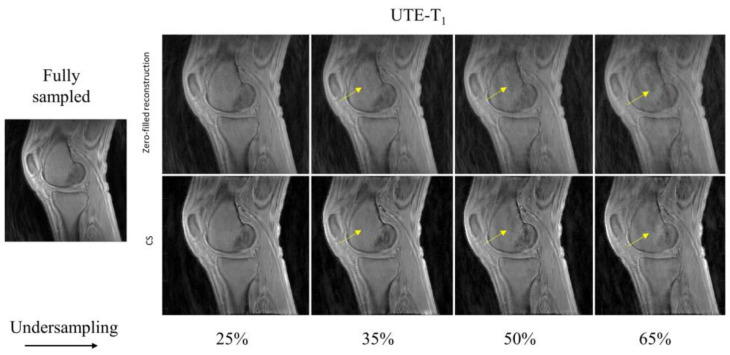
Comparison of UTE-T_1_-reconstructed images at different undersampling rates between zero-filled reconstruction and CS-based reconstruction. Reference image on the left was reconstructed using fully sampled k-space data. Yellow arrows indicate reconstruction error prevalent on zero-filled reconstruction (**top row**) and the reduced artifacts in the CS-based reconstruction (**bottom row**).

**Figure 3 sensors-22-07459-f003:**
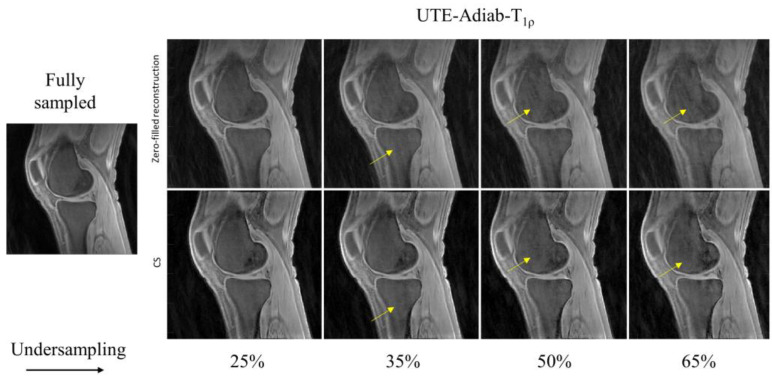
Comparison of UTE-Adiab-T_1ρ_-reconstructed images at different undersampling rates between zero-filled reconstruction (**top row**) and CS-based reconstruction (**bottom row**). Image reconstructed using entire k-space data is shown as the reference. Reconstructed images using incomplete (i.e., undersampled) k-space via zero-filling carries streaking artifacts that are resolved using CS-based reconstruction, as marked by the yellow arrows.

**Figure 4 sensors-22-07459-f004:**
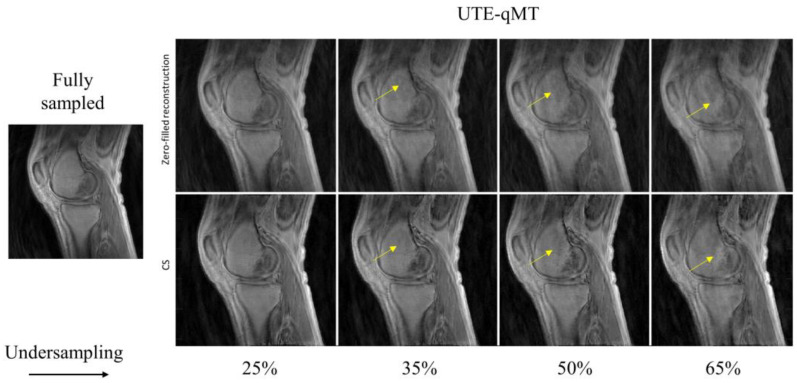
Comparison between zero-filling-based reconstruction (**top row**) and CS-based reconstruction (**bottom row**) for UTE-qMT at different undersampling rates (25%, 35%, 50%, and 65%). The arrows point towards the streaking artifacts and blurring present in the reconstructed images with zero-filling, which is reduced with CS.

**Figure 5 sensors-22-07459-f005:**
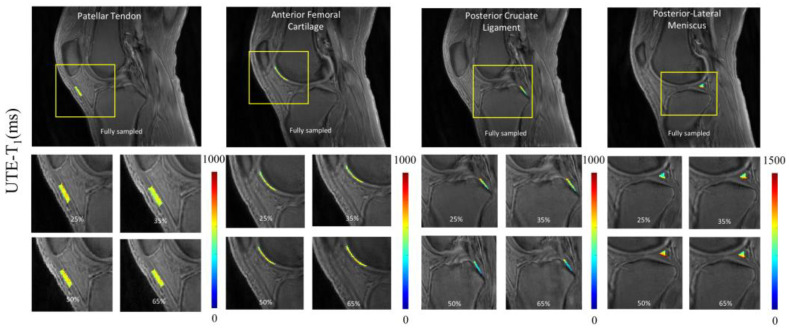
Quantitative mapping of UTE-T_1_ parameter at different undersampling levels in four representative ROIs. A representative region is chosen from each group, namely tendon (patellar tendon), cartilage (anterior femoral cartilage), ligament (posterior cruciate ligament), and meniscus (posterior lateral meniscus). The T_1_ quantification values ranged between 500 to 1000 ms. The parameter maps show high correlation between different undersampled maps except posterior lateral meniscus, with values between 900 and 800, a margin of 100 ms.

**Figure 6 sensors-22-07459-f006:**
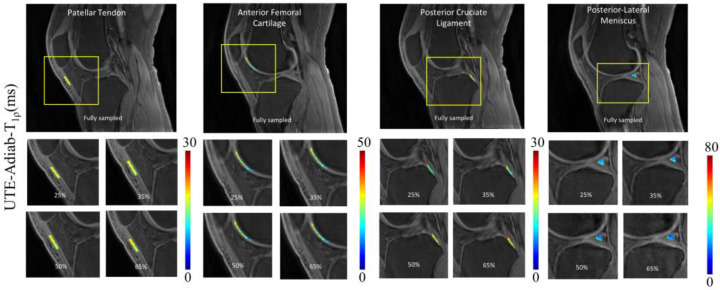
Quantitative mapping of UTE-Adiab-T_1ρ_ parameter at different undersampling levels in four representative ROIs. The estimated T_1ρ_ parameter ranged between 15 and 40 ms in the aforementioned ROIs.

**Figure 7 sensors-22-07459-f007:**
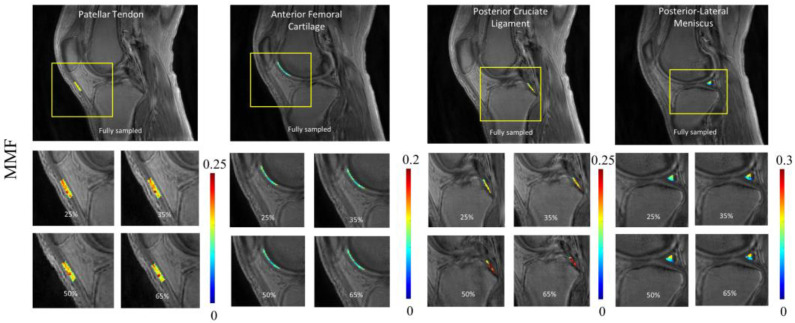
Quantitative mapping of UTE-qMT parameter (i.e., MMF) at different undersampling levels in four representative ROIs. The estimated MMF ranged between 0.12 and 0.19. For the PCL region, due to partial volume effect, quantitative maps showed visible changes between images with different undersampling rates.

**Figure 8 sensors-22-07459-f008:**
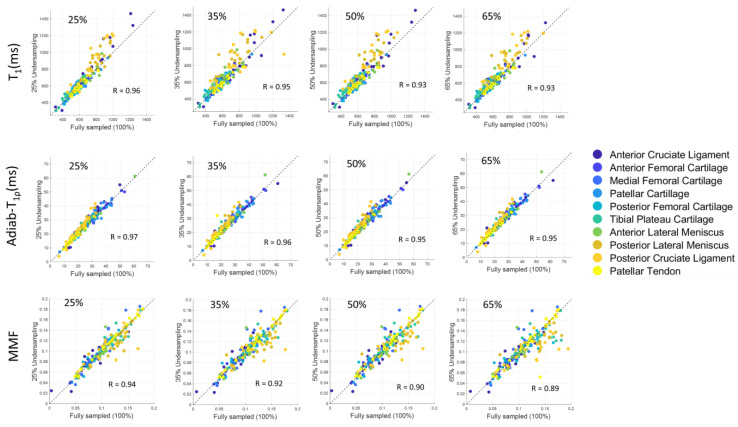
Scatter plot for various quantification parameters (i.e., T_1_, T_1ρ_, and MMF) at different undersampling levels in ROIs from human knee joint. Correlation coefficient (R) recorded over 0.85 for all parameters in all four undersampling rates. While very few points can be recognized as outliers, parameters estimated in most of the ROIs showed high linearity between undersampled CS reconstruction and fully sampled reconstruction.

**Figure 9 sensors-22-07459-f009:**
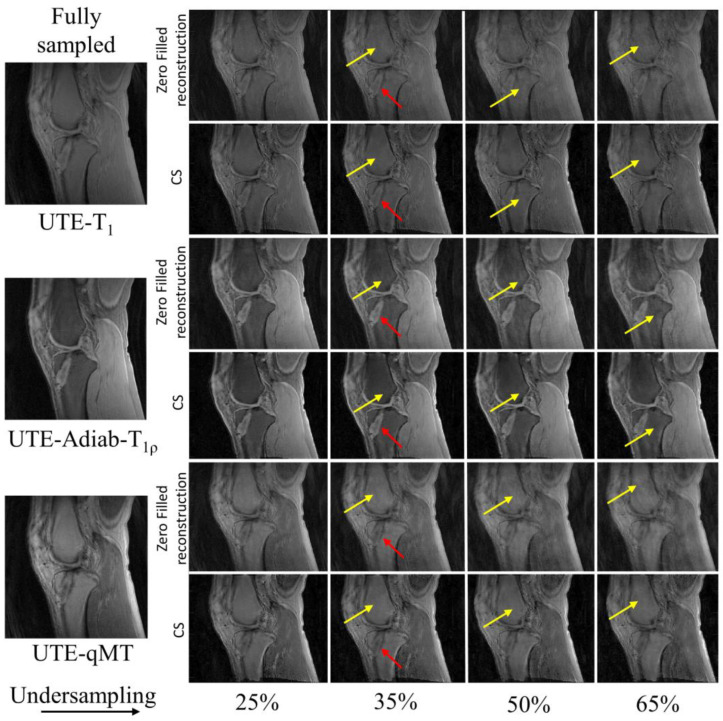
CS reconstruction in a patient with a degenerated knee. As seen, the presence of an abnormality (tibial tunnel from an ACL reconstruction as shown by red arrows) is clearly depicted in images with CS reconstruction. Streaking artifacts in zero-filled reconstruction are dramatically reduced in CS reconstruction (yellow arrows).

**Table 1 sensors-22-07459-t001:** This table provides all the parameters used in VFA-UTE-T_1_ [7], UTE-Adiab-T_1ρ_ [12], and UTE-qMT [15]. Additional B_1_ correction was performed using actual flip angle imaging (AFI) technique [28]. For qMT, a Fermi pulse was used for MT saturation.

	Sequence	UTE—T_1_	UTE—Adiab T_1ρ_	UTE—qMT
Parameters				
**TR (ms)**		20	500	102.6
**TE (ms)**		0.032	0.032	0.032
**FOV (cm^3^)**		15 × 15 × 10.8	15 × 15 × 10.8	15 × 15 × 10.8
**MT power (°)**		-	-	500, 1500
**Frequency** **offset (kHz)**		-	-	2, 5, 10, 20, 50
**Spin lock time (ms)**		-	0, 24, 48, 96	-
**Flip angle (°)**		5, 15, 30	10	7
**Matrix**		256 × 256 × 32	256 × 256 × 32	256 × 256 × 32
**Receiver bandwidth (kHz)**		166	166	166
**Scan time**		6 min 8 s	8 min 54 s	9 min 38 s

**Table 2 sensors-22-07459-t002:** This comparison of zero-filled (ZF) and CS reconstruction using MSSIM measure at various undersampling rates. MSSIM values for various flip angles of UTE-T_1_ show that with increase in undersampling rate, MSSIM falls below 0.9 for CS images and much lower value (~0.7) for ZF-reconstructed images. For MSSIM metric value of UTE-Adiab-T_1ρ_ at different spin lock times, an undersampling rate of 65% still yields significantly higher MSSIM measure (around 0.95), which proves the efficiency of CS-based reconstruction, whereas ZF-reconstructed images average at 0.8. In UTE-qMT with two different MT powers (1500 and 500 degree) at multiple frequency offsets (2, 5, 10, 20, and 50 kHz) a mean of 0.93 at 50% undersampling has been recorded, proving the CS reconstruction strategy to be quite more reliable than ZF. The recorded MSSIM for CS is closer to fully sampled data (<10% error) than ZF (approx. 30% error).

		ZF-Based Reconstruction MSSIM					CS-Based Reconstruction MSSIM				
	Sequence Parameter	Fully Sampled	25	35	50	65	Fully Sampled	25	35	50	65
UTE-T_1_	FA = 5	1.00	0.88	0.84	0.76	0.68	1.00	0.99	0.98	0.93	0.88
	FA = 15	1.00	0.88	0.84	0.76	0.68	1.00	0.97	0.94	0.89	0.84
	FA = 30	1.00	0.88	0.84	0.76	0.69	1.00	0.97	0.93	0.89	0.84
UTE-Adiab-T_1ρ_	TSL = 0	1.00	0.89	0.85	0.77	0.71	1.00	0.99	0.99	0.97	0.94
	TSL = 24	1.00	0.88	0.85	0.77	0.72	1.00	1.00	0.99	0.98	0.97
	TSL = 48	1.00	0.89	0.85	0.76	0.72	1.00	0.99	0.99	0.98	0.97
	TSL = 96	1.00	0.88	0.85	0.76	0.72	1.00	1.00	0.99	0.98	0.96
UTE-qMT w/MT Power 1500°	Offset = 2	1.00	0.89	0.84	0.77	0.69	1.00	0.99	0.98	0.93	0.87
	Offset = 5	1.00	0.89	0.84	0.77	0.69	1.00	0.99	0.98	0.92	0.87
	Offset = 10	1.00	0.89	0.84	0.77	0.69	1.00	0.99	0.97	0.93	0.91
	Offset = 20	1.00	0.89	0.84	0.76	0.69	1.00	0.99	0.98	0.93	0.87
	Offset = 50	1.00	0.89	0.84	0.76	0.69	1.00	0.99	0.97	0.93	0.87
UTE-qMT w/MT Power 500°	Offset = 2	1.00	0.89	0.84	0.76	0.69	1.00	0.99	0.97	0.93	0.87
	Offset = 5	1.00	0.89	0.84	0.76	0.69	1.00	0.99	0.97	0.93	0.87
	Offset = 10	1.00	0.89	0.84	0.77	0.7	1.00	0.99	0.98	0.94	0.89
	Offset = 20	1.00	0.89	0.84	0.77	0.7	1.00	0.99	0.98	0.93	0.88
	Offset = 50	1.00	0.89	0.84	0.77	0.7	1.00	0.99	0.98	0.94	0.88

**Table 3 sensors-22-07459-t003:** Parameter quantification for 10 ROIs with mean and standard deviation along with percentage error. A total of 20 subjects were evaluated for the current study. Error estimates include either an underestimation or overestimation from reference value denoted by the sign. Abbreviations of each ROI are presented on the left side of table. Mean error can be estimated as less than 5% with exceptions in PCL, which is a difficult region to quantify due to the small ROI size and strong partial volume effect through planes (ACL = anterior cruciate ligament, AFC = anterior femoral cartilage, MFC = medial femoral cartilage, PC = patellar cartilage, PFC = posterior femoral cartilage, TPC = tibial plateau cartilage, ALM = anterior lateral meniscus, PLM = posterior lateral meniscus, PCL = posterior cruciate ligament, and PT = patellar tendon).

	Undersampling (%)	ACL	AFC	MFC	PC	PFC	TPC	ALM	PLM	PCL	PT
T_1_ (ms)	Fully sampled	822.8 ± 103.6	584.1 ± 73.9	553.5 ± 118.7	605.1 ± 102.5	557.1 ± 80.5	501.4 ± 93.1	690.3 ± 104	894.5 ± 101.3	838.7 ± 103.7	582.6 ± 81.2
	25	801.6 ± 145.7 (2.6%)	598.5 ± 75.6 (−2.5%)	536.8 ± 89.7 (3%)	631.3 ± 121.8 (−4.3%)	534.2 ± 71.5 (4.1%)	480.8 ± 87.5 (4.1%)	683.6 ± 90.7 (1%)	791.7 ± 100.2 (1.5%)	766.5 ± 104.5 (8.6%)	589.2 ± 87.5 (−1.1%)
	35	783.9 ± 160.4 (4.7%)	598.9 ± 76.8 (−2.5%)	531.8 ± 87.8 (3.9%)	633.9 ± 125.6 (−4.8%)	521.3 ± 76.5 (6.4%)	478.4 ± 94 (4.6%)	669.3 ± 92.3 (3%)	749.2 ± 103.2 (4.6%)	755.7 ± 100.5 (9.9%)	590.9 ± 88.2 (−1.4%)
	50	779.4 ± 132.4 (5.3%)	601.3 ± 76.6 (−2.9%)	528.5 ± 94.3 (4.5%)	635.5 ± 120.4 (−5%)	521.3 ± 76.1 (6.4%)	474.5 ± 94.9 (5.4%)	669.6 ± 95.2 3.0%)	744.8 ± 105.6 (5.6%)	726.7 ± 98.5 (13.4%)	592 ± 86.7 (−1.6%)
	65	775.5 ± 125.6 (5.7%)	604 ± 78.2 (−3.4%)	526.5 ± 94 (4.9%)	637.8 ± 129.2 (−5.4%)	519.9 ± 78.2 (6.7%)	466.8 ± 92.2 (6.4%)	667.1 ± 96.4 (3.4%)	770.1 ± 108.5 (6.2%)	722.8 ± 103.5 (13.8%)	593.8 ± 87.1 (−1.9%)
Adiab-T_1ρ_ (ms)	Fully sampled	29.0 ± 9.2	36.0 ± 8.4	27.0 ± 7.4	29.2 ± 5.2	25.7 ± 6.5	22.0 ± 6.5	27.2 ± 5.7	25.2 ± 5.1	21.7 ± 6.7	16.3 ± 5.6
	25	28.2 ± 11.4 (2.7%)	35.9 ± 8.4 (0.2%)	25.6 ± 6.4 (5.2%)	29.5 ± 5.6 (−0.9%)	24.3 ± 6.1 (5.4%)	21.1 ± 5.5 (4.1%)	26.6 ± 5.6 (2.2%)	23.8 ± 5.6 (5.6%)	19.8 ± 6.4 (8.9%)	16.3 ± 5.7 (0.2%)
	35	28.1 ± 10.4 (3.1%)	36.1 ± 8.5 (−0.3)	25.4 ± 6.2 (5.9%)	29.5 ± 7.6 (−0.9%)	24.0 ± 6 (6.6%)	20.9 ± 5.8 (5.0%)	26.5 ± 5.5 (2.5%)	23.1 ± 5.4 (8.3%)	19.5 ± 6.5 (10.3%)	16.2 ± 5.6 (0.8%)
	50	28 ± 11.4 (3.3%)	35.8 ± 8.4 (0.5%)	25.2 ± 6.5 (6.7%)	29.6 ± 5.4 (−1.2%)	23.9 ± 6.1 (6.9%)	20.9 ± 5.4 (5.0%)	26.2 ± 5.8 (3.6%)	22.5 ± 5.4 (10.8%)	19.3 ± 6.8 (11.2%)	16.2 ± 5.7 (0.8%)
	65	27.1 ± 11.1 (6.5%)	36.4 ± 8.5 (−1.2%)	25.2 ± 6.8 (6.7%)	29.6 ± 5.6 (−1.2%)	23.7 ± 6 (7.7%)	20.7 ± 6.2 (6.0%)	26 ± 5.6 (4.4%)	22.5 ± 5.8 (10.8%)	18.9 ± 6.8 (13.1%)	16.0 ± 5.8 (1.8%)
MMF (%)	Fully sampled	8.6 ± 3.0	7.9 ± 1.5	10.9 ± 3.8	9.8 ± 2.2	10.3 ± 2.2	12.7 ± 2.2	10.2 ± 2.4	12 ± 2.3	10.9 ± 2.4	11.9 ± 3.1
	25	8.6 ± 3.3 (0%)	7.8 ± 1.5 (1.3%)	10.7 ± 2.7 (1.8%)	9.8 ± 2.4 (0%)	10.2 ± 2.4 (1%)	12.7 ± 2.2 (0%)	10.2 ± 2.4 (0%)	12.2 ± 2.3 (−1.7%)	11.0 ± 2.8 (−1%)	11.7 ± 3.1 (1.7%)
	35	8.4 ± 3 (2.3%)	7.8 ± 1.5 (1.3%)	10.7 ± 2.7 (1.8%)	9.9 ± 2.6 (−1%)	10.2 ± 2.5 (1%)	12.8 ± 2.2 (−0.8%)	10.3 ± 2.3 (−1%)	12.3 ± 2.3 (−2.5%)	11.2 ± 2.8 (−2.7%)	11.6 ± 3.2 (2.5%)
	50	8.3 ± 3 (3.5%)	7.8 ± 1.5 (1.3%)	10.6 ± 2.6 (2.8%)	9.9 ± 2.6 (−1%)	10.1 ± 2.6 (1.9%)	12.8 ± 2.2 (−0.8%)	10.4 ± 2.3 (−2%)	12.3 ± 2.4 (−2.5%)	11.2 ± 2.8 (−2.7%)	11.6 ± 3.2 (2.5%)
	65	8.3 ± 2.8 (3.5%)	7.7 ± 1.5 (2.5%)	10.6 ± 2.6 (2.8%)	9.9 ± 2.6 (−1%)	10.7 ± 2.6 (−3%)	13 ± 2.2 (−2.3%)	10.4 ± 2.3 (−2%)	12.5 ± 2.5 (−4.1%)	11.2 ± 2.8 (−2.7%)	12.2 ± 3.3 (−2.5%)

## Data Availability

The deidentified data and the codes related to this study are available from the corresponding author upon reasonable request.

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
