# Peer review of "Accelerated Quantitative 3D UTE-Cones Imaging Using Compressed Sensing"

_sensors, 2022, doi:10.3390/s22197459_

Round 1

Reviewer 1 Report

The paper performed an experiment to evaluate the feasibility of using qUTE-Cone with compressed sensing to reconstruct knee MRI images. In particular, 20 subjects were recruited and parameters for image reconstructions are estimated and maintained for all the cases.

The quality of reconstructed images in terms of SSIMs and visual quality are compared at various setttings. Particularly, the paper presented an interesting result when the techniques were used in subjects on conditions in Figure 9.

However, the following points are minor issues that shall be revised. 

- The contribution of the work is not clearly identified in the introduction.

- In Figures 4 and 9, a better presentation of streaking artifacts on images shall be provided.

- In figure 8, the label of the x-axis shall be denoted and the implication of the result (high linearity) shall be discussed.

Author Response

Dear Reviewer,

Thank you for the valuable comments and suggestions. We have addressed the issues to the best of our ability. Please find our responses in the document attached herewith.

Reviewer 2 Report

The AUs describe an accelerated quantitative 3D ultrashort echo time cones imaging technique that utilizes compressed sensing. The MS has merit to be a part of this special issue, however, many changes are suggested to help with the presentation and make it more understandable to the readers. 

The AUs and lab are well established and known for their fast imaging work.   However, the choice of some of the introductory MRI language gives the reader the impression that the MS was written by a non-MRI scientist.  The AUs should be careful in their choice of language.

L30       It is inappropriate to state that MRI has non-radioactive properties.  It is better to state that MRI utilizes non-ionizing radiation.

L32       Longer scan time or slower scan rate but not slower scan time.

L35       … in pain.  not … under pain.

L37       … help reduce healthcare…, not help in reducing healthcare…

L38       It is not efficient patient scheduling, but higher patient throughput.

L40       Stating tissues with short T2 components implies that there are other components with larger relaxation times, which would be imageable.  Values is a better word than components, or consider rewording so it in not ambiguous.

The AUs should be consistent in their use of commas in lists.  They have used “A, B, and C” and “A, B and C” constructions.  My preference is the former. 

L87       No indent

L90       No initial comma, it is after equation (2) on the previous line.

L102     Missing article:  … an FID signal…

L110     Why are there 3 different colored cones but only two different colored k-space trajectories?

Table 2 MMF(%)

L100     I do not find the 9 lines of the materials and methods section sufficient to explain to the familiar or unfamiliar reader what is going on in the sequences.  My advice is to explain things properly so all can understand it, or leave it out completely and assume the MS is for the much narrower audience familiar with the lab’s research.

L204     The details discussed in the MS are difficult to see in the images by an untrained viewer.  I suggest the AUs consider a different presentation.  Perhaps difference images.  (i.e., differences between the fully sampled and under sampled images.)  This should better accentuate differences between the images.

Note to AUs and MDPI:

*    Fig 8 is too small to be read.  Fig 9 is too small to compare parts.  ­­It would be nice to have the option of clicking on a small figure and seeing an enlarged version. 

*    Palatino Linotype has two issues. 

      1.) The font has trouble with the lower case Greek symbol rho. 

      2.) Subscripts are not true subscripts but just a smaller font.

Author Response

Dear Reviewer,

Thank you for reviewing our article and providing us with valuable comments and corrections. We have attempted our best to respond to all the concerns raised. Please find the responses in the attached document. 

Reviewer 3 Report

The authors test the feasibility of Parallel Imaging (PI) combined
with Compressed Sensing (CS, wavelet transform) based on ultra short
echo time (UTE) imaging acquisition. The manuscript is well written,
accessible and mostly sound.

I am a little confused regarding the incorporation of parallel
imaging. The authors say that this study is using PI but they do not
mention it the discussion or conclusion. Perhaps the reader would
appreciate more clarity on that.     

I recommend to remove all data and discussions related to zero
filling. It is obvious that compressed sensing is superior. It
distracts from the main focus to study and compare different
undersampling levels and clutters up figure panels for no gain.

Reasonable correlation of qUTE parameters obtained by CS as compared
to fully sampled. Do the authors have any guess why correlations for
T1_rho appear to be much better as compared to MMF and T1?

Some minor comments are below.

- proofread references. Random numbers occur in names

- Sec 2.1 is very brief. That is o.k. if the authors provide
  references for further reading, relevant to the various protocols
  used by the authors.

- Line 151 to 153 is a repetition of lines around Eq. 1 since symbols
  have been introduced already.

- Eq. 1 at page 4 should be labeled Eq. 3

- Eq. 1 at page 5 should be labeled Eq. 4

Author Response

Dear Reviewer,

Thank you for accepting and reviewing our manuscript. We have tried our best to address the comments and queries raised. Please find attached the responses for all the questions and concerns. 

Reviewer 4 Report

In this study the authors investigate feasibility of qUTE-Cones sequences for quantitative determination of various sample properties, such as T1, T1_rho, macromolecular fraction (MMF) by magnetization transfer. The study was performed on healthy volunteers who had knee imaged by this sequence with complete k-space data acquisition. This data was then retrospectively under sampled with different rates to reconstruct the corresponding compressed sensing images (CS) and zero-filled images. Performance of different undersampling and reconstruction strategies were compared in different anatomical regions of the knee. They found low errors in images with up to 50% undersampling in CS reconstruction and good correlation with fully sampled images.

This manuscript is well written for someone who is already very familiar with the field, a specialist for compressed sensing and MR imaging with ultra-short echo times. However, to the average reader of Sensors, the subject covered here is probably unfamiliar and needs further explanation. Other than that, the manuscript is well organized and results are clearly presented. I recommend acceptance of the manuscript after its revision addressing the following comments.

General comments

1.     Explain in more detail k-space trajectories in UTE. For example, why is used 3D spiral and not radial acquisition? What is its advantage? Why cones; does individual spiral trajectory lie in a cone? How many such spirals are in one cone?

2.     When performing undersampling, were excluded sampled points selected randomly? In this case each spiral would have some excluded points, however, the scan time would not be any shorter as its does depend on the number of sampled spirals and not on the number of points scanned on the spiral as long as its shape remain unchanged. To save scan time, I would expect that entire spirals are excluded. Explain some more details on the used undersampling strategy.

Specific comments

3.     Page 3, line 119, Is there any particular reason for almost twice as many females than males in the cohort?

4.     Page 4, lines 131-141, I suggest that you present sequence parameters in a table, e.g. one column for each sequence and one row for each parameter (TR, TE, FOV …). In addition, I propose that you explain the origin of these sequences, i.e. are they part of a commercially available sequence packages of the used scanner or they were implemented on the scanner by the authors.

5.     Page 4, equation 1, line 149, “CS reconstruction with penalties based on l1 norm was posed as below”, If I am right, this equation in a combination of L1 (second term) and L2 (first term) norms. Is the penalty term then only the second term?

6.     Page 9, Table 2, Digits after the decimal point in standard errors are printed in the next row which makes the table difficult to read.

7.     Page 10, Figure 8, I suggest to add above each column of scatter plots a label for the percentage of undersampling. This important information is now given in vertical axis titles in a relatively small font.

8.     Page 12, lines 338-334, Quality of CS reconstructed images depends on the regularization parameter. You are mentioning here that this parameter was optimized for knee images obtained by qUTE sequence. You can discuss some more on how much different would this parameter be for other anatomical region or if CS would be used with other type of imaging sequences. Which of these two would have a bigger influence on the optimal regularization parameter?

Author Response

Dear Reviewer,

Thank you for reviewing our manuscript and providing valuable comments and suggestions. We have attempted our best to address each concern and query. Please find the responses in the attached word document. 

Round 2

Reviewer 2 Report

It is very confusing to follow corrected corrections (Blue and Red) in a MS.  Please just one set of corrections.  I will mention this again:  The figures are too small and of low resolution to see detail!

L31      …no risk to of exposure to ionizing radiation…

L80      In this study, the feasibility…

L81      This is a very strange, confusing, and possibly redundant sentence construction.  “…for the first time, which has never been demonstrated in literatures to our best knowledge.”  Consider rewriting it as, “In this first of its kind, previously unpublished study, the feasibility and efficacy of PI based CS are explored in 3D qUTE-Cones imaging.”

L114    Is the RF pulse square in the time or frequency domain?  Is it slice or volume selective, or neither?

L118    … in each Cone, yieldsing ~10000-40000 spokes to …

L120    …by applying an additional pulse sequence module designed to achieve the desired signal contrast in front of the UTE-Cones sequence

L154    imaging parameters as shown listed in tTable 1.

L334    The high degree of linearity indicates that the quantitative parameters from undersampled data (y-axis in scatter plots in Figure 8) correlated well with the corresponding quantitative parameters from fully sampled data (x-axis in scatter plots in Figure 8), despite the reduced data.

L343    That I the correlation coefficient R?  Is this the same as R2?

Author Response

Thank you for the reviews. The authors have incorporated all changes from the reviewer's suggestions. Please find attached the response.
